# Factors influencing healthcare workers' and health system preparedness for the COVID-19 pandemic: A qualitative study in Ghana

Osamuedeme J. Odiase[1]*, Akua O. Gyamerah[2], Fabian Achana[3], Monica Getahun[1], Clara Yang[4], Sunita Bohara[4], Raymond Aborigo[3], Jerry John Nutor[5], Hawa Malechi[6], Benedicta Arhinful[7], John Koku Awoonor-Williams[8], Patience A. Afulani[1,9]

1 Institute for Global Health Sciences, University of California, San Francisco, California, United States of America, 2 Department of Community Health and Health Behavior, University at Buffalo, Buffalo, New York, United States of America, 3 Navrongo Health Research Centre, Navrongo, Ghana, 4 University of California Berkeley, Berkeley, California, United States of America, 5 Department of Family Health Care Nursing, University of California, San Francisco, California, United States of America, 6 Tamale Teaching Hospital, Tamale, Ghana, 7 Johns Hopkins Bloomberg School of Public Health, Baltimore, Maryland, United States of America, 8 Formerly Policy Planning Monitoring and Evaluation Division, Ghana Health Service, Accra, Ghana, 9 Department of Epidemiology & Biostatistics, University of California, San Francisco, California, United States of America

* Osamuedeme.odiase@ucsf.edu

**Data Availability Statement:** The data used for the analysis is included with the submission as a supplementary file.

## Abstract

Adequate preparedness of health systems, particularly healthcare workers (HCWs), to respond to COVID-19 is critical for the effective control of the virus, especially in low- and middle-income countries where health systems are overburdened. We examined Ghanaian HCWs' perceived preparedness to respond to the pandemic and the factors that shaped their preparedness and that of the health system. Semi-structured in-depth interviews were conducted with n = 26 HCWs responsible for the clinical management of COVID-19 patients and three administrators responsible for developing and implementing COVID-19 policies at the facility level. Interviews were conducted over the phone in English, transcribed, and analyzed using a thematic analysis approach. Generally, HCWs felt inadequately prepared to contain the spread of COVID-19 due to resource shortages and inadequate training. HCWs, similarly, perceived the health system to be unprepared due to insufficient clinical infrastructure and logistical challenges. The few who felt prepared identified readiness in managing high consequence infectious disease cases and pre-existing protocols as enablers of HCW preparedness. The health system and HCWs were unprepared to manage the COVID-19 pandemic due to inadequate training, logistical challenges, and weak clinical infrastructure. Interventions are urgently needed to improve the health system's preparedness for future pandemics.

## Introduction

On March 11, 2020, the World Health Organization (WHO) declared the COVID-19 outbreak a global pandemic [1], several months after a cluster of severe viral pneumonia cases were

**Funding:** This work was funded by the University of California, San Francisco COVID-19 Related Rapid Research Pilot Initiative in the form of a grant (#2016796) received by PAA. The funders had no role in study design, data collection and analysis, decision to publish, or preparation of the manuscript. No additional external funding was received for this study.

**Competing interests:** The authors have declared that no competing interests exist.

reported in Wuhan, China [2]. Since then, over 775 million people have contracted the virus, and over 7 million people have succumbed to COVID-19 globally [3]. The surge in COVID-19 cases during the early phase of the pandemic threatened to overwhelm the capacity of health systems and overburden healthcare workers (HCWs). The pandemic unveiled HCWs' insufficient preparedness to effectively manage patients while also safeguarding their own health. Globally, many HCWs were disproportionately vulnerable to COVID-19 because of their role as frontline workers, with nearly 1.4 million (10%) of all COVID-19 cases documented among HCWs by July 2020 [4]. Further, between January 2020 and June 2021, 80,000 to 180,000 HCWs died as a result of COVID-19 [5].

Global health security, including the ability of developing countries to respond to emerging pandemics such as COVID-19, is a public health priority [6]. The pandemic, however, placed an unprecedented burden on the already overstretched health systems in Africa. Furthermore, of the more than 150,000 COVID-19 cases among HCWs in Africa, approximately 70% of these were in Algeria, Ghana, Kenya, South Africa, and Zimbabwe [7]. The loss of frontline HCWs due to COVID-19 has had significant consequences for service delivery, putting HCWs and health systems in perilous positions [7].

Adequate preparedness of HCWs is critical for managing COVID-19. Yet, evidence from several studies highlight low perceived preparedness among HCWs [8–11]. A cross-sectional study conducted in Ethiopia found that nearly three-fourths of HCWs felt anxious treating febrile patients [10]. Similarly, a study in Libya found a majority of HCWs (77.1%) were not prepared to respond to the pandemic, with 54% of HCWs citing lack of training in using personal protective equipment (PPEs) [11]. In particular, global supply chain shortages of PPEs early in the pandemic contributed to the high infection and mortality rate among HCWs [12]. Other documented factors contributing to lack of preparedness include inadequate knowledge of case definition, inability to recognize high-risk patients, lack of tests to identify suspected cases, lack of protocols, and poor clinical infrastructure [8,13,14].

African nations had among the lowest scores on indicators of their preparedness to respond to public health emergencies [6,15]. Limited financial investment, insufficient budgetary allocation, poor governance, and poor clinical infrastructure have made managing the COVID-19 pandemic challenging for HCWs in Africa [16,17]. Yet, few studies have examined the preparedness of health systems and HCWs in Africa to manage the COVID-19 pandemic from the perspective of HCWs. A scoping review of health systems preparedness in Africa found that testing capacity in most African countries was low compared to their population sizes, which seriously undermines the preparedness of health systems to effectively respond to pandemics. The health system in Ghana has faced numerous challenges in responding to the pandemic. Ghana has a median of four ventilators and five staffed intensive care unit (ICU) beds, which are unequally distributed across the nation for a population of roughly 30 million [18,19]. When the first COVID-19 cases were confirmed, 10 of the16 administrative regions in the nation, home to a combined population of 10 million people, lacked an ICU bed [18,19]. Even though Ghanaians' health-seeking behaviors were intended to be improved by the introduction of the government-funded national insurance scheme, both public and private facilities have found it difficult to supply the essential resources for their operationalization due to lack of funding and poor budgetary allocation [19]. For instance, on March 11, 2020, the president of Ghana announced an increase of $100 million to the health system [19,20]. Though the funds were intended for equipment purchases, health education, and infrastructure, there were difficulties in allocating these funds, and basic medical supplies like PPEs were not supplied to HCWs [19].

In Ghana, HCWs' inadequate preparedness has hamstrung the health system and its workers to adequately manage the COVID-19 pandemic [21]. Our quantitative study of HCWs in

Ghana found that HCWs were inadequately prepared to respond to the COVID-19 pandemic and that perceived inadequate preparedness was associated with stress and burnout [8,9]. This study extends the evidence on HCWs and health system preparedness to respond to the pandemic using qualitative data for a more nuanced and in-depth understanding of the issue, to inform future pandemic response strategies. Our aim is to explore HCWs and health system leaders' perceptions of factors that shaped their own preparedness as well as that of the health system.

## Materials and methods

### Setting

On March 12, 2020, Ghana, a West African nation of about 30 million, reported its first two COVID-19 cases [22]. As of June 16, 2024, there have been 172,037 cases and 1,462 COVID-related deaths [23]. COVID-19 vaccination began in Ghana on March 2, 2021 [24], with 35% of the population fully vaccinated as of December 31, 2023 [23]. Ghana is among the top five countries that accounted for 70% of reported COVID-19 infections among HCWs in Africa [7]. Further, Ghana's healthcare system is understaffed, with an estimated 0.1 medical doctors and 2.7 nurses and midwives per 1,000 people [25,26] and 0.5 ICU beds per 100,000 people [18]. The high number of infections combined with a fragile health system heightened HCWs' concerns about their safety and the potentially devastating consequences of the pandemic [27].

### Parent study

This qualitative study was embedded in a larger mixed-methods study of HCWs in Ghana conducted between April 17, 2020, and March 8, 2021. Thus, most of the data were collected in the early phase of the pandemic before vaccines were available in the country. The parent study electronically recruited HCWs nationwide via social media platforms (Facebook, WhatsApp, and direct messaging). Respondents were invited to complete a self-administered online survey. We also utilized healthcare worker professional networks, as well as the Ghana Health Service, to disseminate the survey link via email. Prior to the start of the survey, HCWs were given brief consent language and the option of skipping questions. The self-administered survey, conducted in English, included demographic, perceived preparedness, stress, and burnout questions, as well as others that were relevant to the pandemic response. Perceived preparedness was assessed in the quantitative study using a 15-item scale capturing personal, facility, and psychological preparedness for prevention, diagnoses, management, and education regarding COVID-19. Each question had response options ranging from 0 (not prepared at all) to 3 (very prepared). A perceived preparedness score was generated from the sum of responses to the 15 questions and categorized based on the mean score. Previous manuscripts on HCWs' perceived preparedness contain additional study methods [8,27,28].

### Qualitative sampling

The study used an exploratory descriptive qualitative research design to investigate HCWs' perceptions of their own preparedness and that of the health system [29–31]. HCWs who participated in the survey were asked to provide their contact information if they would be willing to participate in a follow-up qualitative interview. We generated a random sub-sample (n = 44) from the list of HCWs willing to participate in the interviews (n = 646), which was relatively balanced by position, age, years of experience, gender, geographic region, and their responses to the quantitative measures of preparedness, stress, and burnout. In addition, we purposively sampled facility and health system leaders because they played a critical role in the

COVID-19 response. Potential participants were first contacted by phone and given an overview of the study's aims, objectives, and methods. Interviews were then scheduled for a time convenient for the participants. Providers from the list were sequentially invited to the interviews, skipping those no longer interested or unavailable, and stopping when thematic saturation was achieved—i.e., when no new additional theme was being observed during interviews. A total of n = 29 HCWs were interviewed.

### Qualitative data collection

Two interview guides were developed: one for frontline HCWs and one for key informants who held leadership roles in their health facility. Frontline HCWs were asked about their experience in responding to the pandemic: (1) perceptions of their preparedness, (2) factors shaping HCW COVID-19 preparedness, (3) challenges and facilitators of the COVID-19 response, and (4) the impact of COVID-19 on clinical care/experiences and their life and psychological wellbeing. Key informants were asked about: (1) the COVID-19 response at the health system level, (2) health facility policies and procedures, and (3) challenges and facilitators of the response. Both groups were asked about recommendations for the country's COVID-19 response. All interviews, lasting 45 to 60 minutes, were conducted via phone and in English by a researcher with qualitative training (FA).

### Data analysis

Interviews were audio recorded, transcribed, and reviewed by the study team. We utilized a group-based thematic analysis approach [32]. Four researchers (AOG, OO, CY, and FA) developed an initial deductive codebook based on the interview guides, and independently coded transcripts using Dedoose software; three transcripts were collaboratively reviewed to ensure interrater reliability. We compared and discussed coding in detail, iteratively refining the codebook and adding inductive codes. The remaining transcripts were then divided among three researchers (OO, CY, and FA) and coded independently. Codes specific to HCW preparedness and health system preparedness were then queried and analyzed for emergent themes. Analytic summaries of the codes were organized in a thematic table and grouped as follows: 1) adequate HCW preparedness; 2) inadequate HCW preparedness; 3) adequate health system preparedness; 4) inadequate health system preparedness; and 5) recommendations for health system preparedness.

### Ethics approval

We obtained ethics approval from the Institutional Review Boards of the University of California San Francisco (#20–30656) and the Navrongo Health Research Centre Institutional Review Board (#NHRCIRB374) in Ghana. All participants were given electronic consent language and were asked to confirm their consent by taking the survey. Additional verbal consent was obtained for the interviews.

## Results

### Participant characteristics

Table 1 shows the demographic information and other relevant quantitative data for the respondents in the qualitative arm of the study, taken from the survey data collected between April 2020 and May 2020. Of the n = 29 participants, n = 26 were frontline HCWs (n = 10 doctors, n = 13 nurses and midwives, and n = 3 physician assistants) from various government hospitals, teaching hospitals, and private hospitals across Ghana. Three (n = 3) participants

**Table 1. Background characteristics of healthcare workers and administrators.**

| Participant characteristics | N = 29 | % |
|---|---|---|
| **Provider type** | | |
| Doctor | 10 | 34.5 |
| Nurse/medical or physician assistant | 16 | 55.2 |
| Administrator | 3 | 10.3 |
| **Gender** | | |
| Female | 15 | 51.7 |
| Male | 14 | 48.3 |
| **Age (y)** | | |
| Less than 30 | 11 | 37.9 |
| 30 to 39 | 14 | 48.3 |
| 40 to 73 | 4 | 13.8 |
| **Family composition** | | |
| Married with one or two children | 9 | 31.0 |
| Married with three to six children | 6 | 20.7 |
| Single with one or two children | 2 | 6.9 |
| Single with no children | 11 | 37.9 |
| Missing/unknown | 1 | 3.4 |
| **Years of experience** | | |
| 5 or less | 15 | 51.7 |
| 6 to 10 | 9 | 31.0 |
| More than 10 | 5 | 17.2 |
| **Region** | | |
| Greater Accra | 2 | 6.9 |
| Northern | 7 | 24.1 |
| Oti | 2 | 6.9 |
| Upper East | 5 | 17.2 |
| Upper West | 2 | 6.9 |
| Volta | 3 | 10.3 |
| Western | 2 | 6.9 |
| Other (e.g., Ashanti, Bono East, Central, Eastern, North East) | 5 | 17.2 |
| Unknown/preferred not to answer | 1 | 3.4 |
| **Facility type** | | |
| Government facility | 17 | 58.6 |
| Private/mission facility | 5 | 17.2 |
| Teaching hospital | 6 | 20.7 |
| Unknown/preferred not to answer | 1 | 3.4 |
| **Preparedness** | | |
| Not at all prepared | 9 | 31.0 |
| A little prepared | 10 | 34.5 |
| Prepared | 8 | 27.6 |
| Unknown/preferred not to answer | 2 | 6.9 |

were administrators responsible for developing and implementing COVID-19 management policies at their respective facilities. Over half (62%) were 30 years of age or older and about half (52%) were female. Two-thirds (66%) of the respondents in the qualitative arm (n = 19) expressed inadequate preparedness to manage COVID-19, with 31% (n = 9) reporting they did not feel at all prepared, while 35% (n = 10) reported feeling only a little prepared. In the

qualitative interviews, respondents discussed factors shaping their perceived preparedness and made recommendations to improve preparedness for future pandemic responses, which are presented below.

### Factors shaping inadequate preparedness of HCWs and the health system

HCWs who felt unprepared discussed several experiential and structural factors that contributed to their preparedness level and that of the health system, including inadequate training, logistical challenges, and insufficient clinical infrastructure.

**Inadequate training.** Inadequate COVID-19 pandemic preparedness was attributed to inadequate training, particularly a lack of specialized training, as well as delayed training. One respondent, for instance, reported delays in receiving formal training at the onset of the pandemic as contained in the excerpt below.

> "*Facilities should have been sorted out in terms of PPEs and training them on that, training on COVID-19 and education on it and how to use PPEs and availability of PPEs. We had the training two months towards June and people were coming and really making use of the education. We teach you how to use nose masks and those things. If not when it [the pandemic] started around March, people were not coming and we were just there learning on our own, so we didn't start with the education. We didn't start with the training.*"–Doctor HCW 5

Another participant did not anticipate that COVID-19 would affect Ghana, which subsequently contributed to the delayed training of HCWs.

> "*I must say that really when we were hearing about COVID, as a service, we didn't think that COVID was going to come to Ghana. So, the preparations really were late in terms of even training the staff.*"–Other HCW 29

Further, the HCW remarked that the general lack of training in infectious diseases contributed to feelings of unpreparedness:

> "*We need highly trained staff who can deal with infectious diseases, and we don't have that now. If we have that, then our COVID patients, we will just move them there because they already know Infection Prevention and Control (IPC) and all those things. Of course, every staff needs to practice to some level, but we need people who are properly trained so that they can highly perform in the infectious disease unit. So, we need that because we didn't have that and that really was a challenge at the early stages of managing COVID-19.*"–Other HCW 29

Other HCWs noted that lack of refresher trainings, including the absence of trainings focused on the changing trends of the pandemic, contributed to their feeling inadequately prepared:

> "*I said this because of lack of PPEs and refresher training on some of these issues. The refresher training will help you have a clearer understanding about the whole issue, and therefore have more interest. This is so because the trend keeps changing.*"–Nurse HCW 24

**Logistical challenges.** HCWs attributed their inadequate preparedness to the fragmented procurement systems. As one participant noted:

> "*Initial logistics were very few and limited and we did our best [. . .] Procurement processes were slow; things that needed to be bought were not bought in time.*"–Nurse HCW 17

Other participants echoed similar sentiments regarding logistical challenges, noting flaws in the distribution of essential equipment and resources:

"*For the logistics, the distribution didn't go on well. I don't know whether it was supposed to be delivered directly into the departments, which is also not the case because they are giving them to the hospitals. It borders more on management. There are so many things I don't know how to say them but someone saying, "I have given this" and you seeing something different at the wards, those were the issues.*"–Doctor HCW 2

HCWs identified shortages in equipment and other resources needed to manage COVID-19 cases safely and effectively. For example, HCWs reported that there was a lack of PPEs during the onset of the epidemic, which compromised their safety and caused stress. As one respondent explained:

"*The PPEs were not there [...] let's say face masks, for instance, that we were supposed to have, they were limited, and we sometimes have to improvise; we had to improvise with cloth masks [...] In fact, it's stressful. After every four hours, we were supposed to change [the mask] but [the masks] were not there. So, we have to wear [the same mask] throughout the day.*"–Nurse HCW 26

Another HCW revealed that the absence of necessary equipment not only caused fear and worry, but was reported to compromise the quality of care they gave to COVID-19 cases:

"*That is why I am saying the resources that we need were not coming. Even at the time when COVID-19 came there, we were short of nurses, and when you come in contact with a case, we have to quarantine you for some days, and you don't have to come to work. So, the rest has to do extra duty, and so it got to a point due to shortage of staff, even if you come in contact with a case, you still come to work instead of you being quarantined, you continue coming to work and exposing yourself and others and the resources we needed, especially the PPEs, that was my main concern, they were not really there, so we have to depend on cloths and other things just to protect ourselves because of COVID-19. It was just God who protected us*"–Nurse HCW 16

**Information flow challenges.**    HCWs also attributed their lack of preparedness to delays in testing. As a nurse explained:

"*Because, for example, if you have requested for a test and the result come and you are not aware, so you don't even know what, yeah because if you request, if you attend to a case and the case turns out positive, it means that you have been exposed like that.*"–Nurse HCW 26

Additionally, information flow challenges compromised the safety of providers and impacted their ability to effectively manage suspected and confirmed cases. A nurse explained as follows:

"*The person was referred from a facility to my facility and the person has underlying conditions. When the person came to my facility, they told us that the person, they were only managing for asthma and is diabetic, and the person was on inducement. Now with all the things they told us because they said they've done the [COVID-19] testing over there, it wasn't even*

*indicated on the referral that a test was done in that regard. So, we were managing the case as a normal case. As I said, the PPEs were not there; the masks were not even there as in N95 or surgical for you to use, and the person also needs monitoring. Finally, this person, actually, one of our doctors who came and reviewed was saying actually this is the condition, and how the oxygen level was going down and he was like ok, they should do the test, only for them to take the swab and it came out positive [for COVID-19] this time round. Now, how come they missed the diagnosis from that side, and number two, they didn't even inform us that the person was actually tested [. . .] At this time that I am talking, a lot of us have already had contact with the patient. And they are trying to help that person get back her life, so in the long run, about two or three of us were infected, no, about four were infected."*–Nurse HCW 19

**Weak clinical infrastructure.**    HCWs cited insufficient number of isolation centers, ventilators, limited testing capacity, and lack of support staff, as factors that contributed to inadequate preparedness. A respondent noted that existing infrastructures from the Ebola outbreak, were inadequate in responding to COVID-19.

"*Even though we had the Ebola structures that were renovated some time ago, but they couldn't really fit into this because some of those things were quite small. So, these were some of the gaps in the response.*"–Other HCW 29

A participant discussed the limited number of isolation centers:

"*We need more for the isolation center, we had to use one ward for the isolation. We had to leave that ward for the isolation meanwhile our wards are just two. So, the place, if we had so many cases, we would have been in hot waters because our place is very congested. We don't have enough buildings; we need to have facilities, as in buildings so that if there is any case, we can put it there as an isolation center.*–Nurse HCW 11

## Factors shaping adequate preparedness of HCWs and the health system

HCWs reported several experiential and structural factors that contributed to their preparedness and that of the health system. These included adequate PPE availability, knowledge and training, perceived readiness managing high consequence infectious diseases, and pre-existing protocols.

**Sufficient supply of PPEs and adequate knowledge and training.**    Adequate knowledge and training on the pandemic were reported to reduce anxiety and increase provider confidence to manage suspected COVID-19 cases. A participant discussed the quality of the training, including information to dispel some misconceptions:

"*It [the training] was good. It also gave us courage and reduced anxiety. It also gave [providers] much information on misconceptions about the virus, which helped us answer questions asked by our patients.*"–Nurse HCW 14

Similarly, other respondents credited adequate training and sufficient resources:

"*Personally, for me, I'll say I was ready because I had all the resources at my disposal, and through the training I had, I was very confident.*"–Nurse HCW 22

**HCWs' readiness managing high consequence infectious diseases.**    Some providers ascribed their preparedness to their perceived readiness to care for suspected or confirmed

COVID-19 cases, as well as their experience caring for high consequence infectious disease cases. Providers discussed preparedness in the context of past management of COVID-19 cases or other high consequence infectious disease cases:

"*I am prepared because I have been seeing a lot of COVID-19 patients already.*"–Doctor HCW 9

"*Prior to COVID-19, when we were working, we did not know that there's something like COVID-19, but we were always prepared in handling disease conditions whether infectious or non-infectious. And so, you will not prepare for a particular disease like COVID-19, but we always have the universal preparedness that in this field. You have disease conditions that are so dangerous, like HIV; when you get it, you will be on its drugs for the rest of your life. So, any patient that you are approaching, you would always adapt to the universal precautions. COVID-19 only came to add to the numbers but we are always prepared as health workers to handle conditions as they come.*"–Doctor HCW 8

**Pre-existing safety protocols.** HCWs also credited pre-existing protocols, such as hand-washing, for their preparedness, as one participant noted:

"*What is going on well is the observation of the protocols; I'm really happy, even up to now, and for me, my staff, if you are not in mask, I won't even allow you [to work]; you must be in mask before you work. So that one, I take it upon myself as the head. I don't know of other doctors, but for me, because of where we find ourselves, the emergency area, because I always tell them that we do all sorts of consultations so the best is to make sure you are in a mask and we have hand sanitizer and the hand washing bucket so I think on that note we are doing well.*"–Doctor HCW 3

Similarly, HCWs attributed the health system's preparedness to HCWs' adherence to protocols:

"*To me, the checking of temperature at the entrance of the outpatient department (OPD) to me is good; every client coming to the facility wearing the masks is good [. . .] making sure that all relatives and staff wear their masks, because if the health worker doesn't wear [masks], you are endangering yourself, as well as your family members when you go back home.*"–Nurse HCW 20

## Preparedness recommendations

HCWs shared recommendations to improve pandemic preparedness in Ghana. These included the need for better logistical support, sufficient supply of PPEs and ventilators, adequate training on COVID-19 management, improved clinical infrastructure, capacity building, and decentralization of the health system.

**Improved availability of resources and logistical support.** HCWs advised health facilities to have an ample supply of resources:

"*That is what I am saying that the facility also depends on the resources available. If they are not there, what will the staff do? So, it's all about having resources*"–Nurse HCW 16

HCWs also indicated that for the health system to be prepared to respond to the pandemic, logistics need to be considerably improved:

"*I will say that you see when you talk about the PPEs like this, it's when this virus came that we saw most of the PPEs. Sometimes when you see this overall and things, scarcely will you see people wear them, aha. And it was when the virus came that they had to send for some from China and those kinds of things. I think we should have them readily available so that we can also, the health system, will also be [prepared] adequately.*"–Nurse HCW 12

**Adequate training.** HCWs recommended trainings on infectious diseases and psychological preparedness, both of which were perceived to be critical for HCW preparedness. As a participant discussed:

"*So, one of the things we've not been doing even though we're talking about it is the regular training of staff; going through drills and things to go through these things so that when it comes, the staff will be fully prepared psychologically and everything. And again, like we talked about earlier, we need psychological preparedness for all our staff, in every situation that they might encounter and all those things. I think it's very important and it's something we talk about but something we don't do regularly. We only do it when we have epidemic outbreaks like Ebola and then also now COVID. But I think at least it should be something that should be part and parcel of the training of staff and regularly done at our health facilities, regions, and districts.*"–Other HCW 29

HCWs also suggested pandemic preparedness be incorporated into the curricula taught at health institutions to participants:

"*In the training institutions, the health training institutions, that is where we have to start from, so that any person that has an appointment with the Ghana Health Service has already been trained on epidemic outbreaks. So, that when they come, what we only need to do is to add on. But most of these people would come and they have no knowledge about epidemics because we normally will leave that to the medics. We leave that to the technical people. But the nurses and the rest only have knowledge about epidemics. So, right from the training institutions, it should be part of the health curriculum, so that we train them from school before they enter the service. And, when they enter the service, we also give them training.*"–Doctor HCW 1

**Improved clinical infrastructure.** HCWs recommended improving the clinical infrastructure, particularly by increasing the number of isolation centers, testing facilities, and equipment such as ventilators, to ensure health system and HCW readiness:

"*We still have problems with equipment like I mentioned the ventilators, incubators, this no bed syndrome. Somebody was interviewed on TV. I don't know whether you saw her, and she was even a midwife, and they were referring her from a facility in Accra to one facility, but when they called all facilities, there was no bed, so she landed in one of the hospitals. So, enough equipment like beds and all that, so what when people are in an emergency, you can get a bed at least to put the patient.*"–Other HCW 27

**Capacity building.** HCWs noted that capacity building is necessary for the health system to better prepare and respond to pandemics in the future. One participant noted the need for a disaster preparedness program that includes a thorough strategy for resource mobilization, training on COVID-19 management, mental health support, and funding:

"*There should be disaster preparedness programs in place so that we won't wait for something to happen before we will now be looking for ways and means to deal with them, and our staffing level should be enough so that things like these will not run [into a] shortage. We need a lot so that an issue like this we can deal with it and the training was not adequate. Then you come in contact with a case, they quarantine you, nobody cares about how you are feeling, you know. No emotional therapy: you will be scared if the results comes and you are positive. Nobody cares about you there. So, we need to prepare before anything like this so that it will not be surprising, and they should have funds aside for emergencies like.*"–Nurse HCW 16

**Decentralized health system.**   HCWs recommended decentralizing the government response to the pandemic and making information broadly accessible at all levels:

"*The challenges like I said before, especially the national response if we can decentralize it, even to the facility level so that the facility information is available to everyone; okay this is the national situation, this is the regional, this is our facility situation too, so that everybody is aware and has the information.*"–Nurse HCW 26

## Discussion

Our study found that many healthcare workers felt inadequately prepared to respond to the pandemic, specifically noting resource shortages and insufficient training as reasons for their unpreparedness. HCWs also noted that the health system was inadequately prepared due to limited logistics, insufficient clinical infrastructure, and information flow challenges. However, others felt prepared and attributed their preparedness to adequate training and a sufficient supply of PPEs. HCWs also credited pre-existing protocols and perceived readiness to manage high consequence infectious disease cases to their feelings of preparedness.

Our findings are consistent with the broader literature on HCWs' lack of preparedness to combat the COVID-19 pandemic. Though limited, previous studies have found that limited resources such as PPEs and ventilators, contributed to feelings of inadequate preparedness [33]. A qualitative study in Nepal found that lack of PPEs, ventilators, and ICU beds negatively impacted HCW preparedness [34], while a similar study in Yemen showed 86% of HCWs reported insufficient supply of PPEs as a major contributing factor to their lack of preparedness to adequately prevent and control COVID-19 [35]. A study in the Occupied Palestinian Territory found that more than 90% of HCWs did not have PPEs, exacerbating the acute challenges they faced responding to the pandemic [33]. Though failure of the healthcare supply chain contributed to supply shortages worldwide, low- and middle-income countries were disproportionately affected by the lack of resources, particularly reserves of PPEs [36]. HCWs in our study also noted the reuse and potential contamination of PPEs due to supply shortages, echoing similar findings that found the conventional approach of one-use PPEs exacerbated the scarcity of essential equipment and contributed to the lack of preparedness among HCWs [36]. Furthermore, prior studies have found that lack of preparedness contributes to anxiety and depression [37], and as our parent study found, to stress and burnout [27]. The COVID-19 pandemic has underscored the necessity of creating an environment that supports HCWs to be sufficiently prepared to handle future pandemics, regardless of region or health sector [38].

Our study also showed that there were several health system challenges that affected HCWs' perception of preparedness to respond to the COVID-19 pandemic. In particular, similar to other studies, logistical and information flow challenges lowered HCWs' perceived preparedness of the health system [39–42]. An analysis of Ghana's health system revealed that

insufficient supply of adequate equipment is a major factor in the health system's dysfunction [43]. The COVID-19 pandemic has not only caused an unpredictable demand for essential equipment but has also strained already fragile logistic systems; this can potentially explain the low perceived preparedness of the health system, as HCWs do not have sufficient resources to adequately curtail the pandemic. Logistical challenges remain a pressing concern, even among healthcare workers who report adequate preparedness of the health system; this is evidenced by a study conducted in Saudi Arabia [44]. Though HCWs perceived an overall high level of institutional preparedness, many healthcare workers were still deeply concerned about the continuous supply of PPEs [44]. Ineffective management, lack of quality control of PPEs, and insufficient information about availability of PPEs remain an urgent public health concern, highlighting the need to address such issues to improve the health system's preparedness for future pandemics [45–48].

Further compounding the resource challenges, we found that HCWs did not have adequate knowledge, nor did they receive pertinent training to manage COVID-19 cases, particularly early in the pandemic. Our parent study found that 45% of HCWs did not receive training on how to manage the pandemic, and we also found that HCWs with COVID-19 training reported lower levels of stress and burnout [27]. This is concurrent with the global literature, including a quantitative study in the Occupied Palestinian Territory showing that 60% of HCWs did not receive training on how to contain the spread of COVID-19 [33]. The lack of training greatly affected HCW preparedness, as evidenced by only 11.6% of HCWs who felt confident or adequately prepared to address the pandemic [33]. Similarly, a mixed-methods study examining preparedness of HCWs in Uganda's refugee-hosting districts reported that delayed training undermined their HCWs' preparedness, citing critical knowledge gaps related to management of COVID-19 [49]. HCWs had low levels of knowledge on the various treatment options for COVID-19 (56%), appropriate administration of oxygen for COVID-19 patients (42%), and the protocols and recommendations around PPEs (38%) [49]. Similarly, in our parent study, two-thirds (66.4%) of HCWs knew what to do if COVID-19 was suspected, and 35.1% of HCWs reported not knowing how to manage a confirmed case [27]. This could partly be explained by the lack of capacity building [49]. Consequently, HCWs advocated for more capacity building as a means of improving HCW and health system preparedness. Prior studies found that capacity building methods can affect knowledge retention and application of skills [49,50]. Additionally, evidence has shown that capacity building methods such as on-site mentorship and support at point of care can not only improve knowledge retention and application among HCWs, but can also improve patient outcomes [49,51]. For example, a cross-sectional study in Saudi Arabia found that over 90% of HCWs who received training on safe and appropriate usage of PPEs successfully passed the fit test for the N95 respirator mask and most demonstrated fair overall knowledge about the COVID-19 disease [44]. To prepare for future pandemics, HCWs need to receive sufficient training in pandemic preparedness.

Our study found that insufficient clinical infrastructure and lack of human resources contributed to the low perceived preparedness of the health system. Other studies in low-resource settings have reported fragile healthcare infrastructure, particularly insufficient isolation units, ICU beds, and testing centers to adequately screen patients, as factors shaping their ability to effectively respond to the COVID-19 pandemic. A qualitative study conducted in Nepal revealed that the paucity of isolation beds deeply concerned HCWs, with the COVID-19 pandemic only exacerbating the effects of the already fragile clinical infrastructure [34]. This, in turn, contributed to HCWs' perceived low preparedness of the health system. Evidence suggests that India's health system was inadequately prepared to manage the pandemic for much of the same reasons reported by HCWs in our study: insufficient isolation beds and testing delays [52,53]. A study in Kenya evaluated the health system's capacity to effectively manage a

potential surge in COVID-19 cases, and they found significant gaps in the health system's ability to manage a potential surge in cases owing to COVID-19 [54]. Similarly, a cross-sectional study in Ethiopia evaluating the preparedness of HCWs found that HCWs had limited knowledge of COVID-19 detection and management [40]. Dzando et al. [19] conducted a narrative literature review of the challenges faced by the Ghana health system during the COVID-19 pandemic. They found a number of long-standing problems, including lack of human resources, financial constraints, and inadequate clinical infrastructure. This could partly be explained by the fragility of Ghana's health system, impacted by inadequate investments in improving the health infrastructure and building a stronger workforce [55]. It is thus imperative to critically assess present aid strategies as well as governance and coordination structures, and to define new health system strengthening approaches to prepare for future pandemics [38,56].

Though many HCWs expressed inadequate preparedness to manage the pandemic, some attributed their feelings of adequate preparedness to their readiness in managing high consequence infectious disease cases. Therefore, it is probable that HCWs' extensive management of earlier public health emergencies strengthened both their perceived confidence and preparedness to manage the COVID-19 pandemic. This is supported by findings from a cross-sectional study conducted in 57 countries, which showed that HCWs who had no prior outbreak experience scored significantly lower on measures of preparedness compared to HCWs who had prior experience with MERS, SARS, and avian flu outbreaks [57].

## Limitations

This study is a cross-sectional design and therefore limited in assessing the changes in knowledge, training, and response of the health system over time. These findings are relevant to the early phase of the COVID-19 pandemic in Ghana and may not be reflective of later stages and subsequent improvements. However, it is the first study, to our knowledge, to qualitatively explore COVID-19 preparedness among HCWs in Ghana, and serves as a valuable contribution to the literature. Our study is qualitative and therefore not generalizable to the broader HCW population in Ghana. Nevertheless, we included providers from various backgrounds, including varied cadres and positions in the pandemic response. Finally, our methodological rigor, including the team-based analytical approach, inclusion of data collectors in the analysis and interpretation of data, all strengthen our findings.

## Conclusions

Given the immense pressure on HCWs to meet the intensified demand for care, urgent interventions are needed to manage the concerns and needs of HCWs and the health system. The importance of a strong health system, especially during a public health crisis, cannot be overstated. This study provides valuable insight to health facility management and policymakers to address the concerns affecting the preparedness of HCWs and the health system in Ghana. First, Ghana must invest in ensuring PPEs and other critical equipment for the pandemic response are adequately stocked. This will be critical to protecting HCWs and ensuring there aren't labor shortages in an already strained health system. Second, sufficient and ongoing training is needed to prepare HCWs as the pandemic evolves. Third, adequate clinical infrastructure is essential for reducing the risk of an overwhelmed health system. Fourth, capacity building is critical for preparing HCWs to manage the pandemic. Fifth, decentralization of the health system is necessary to maintain optimal functionality of the health system. Our study provides valuable insights into the preparedness of HCWs and the health system during the onset of the COVID-19 pandemic in Ghana and changes needed to strengthen pandemic

preparedness—findings that are informative for policy changes to address gaps and HCWs' preparedness for future pandemic responses.

## Supporting information

**S1 Checklist.**
(DOCX)

**S1 Dataset. Data underlying the study's findings.**
(XLSX)

## Acknowledgments

We would like to thank all the healthcare workers who participated in the study and assisted with survey distribution.

## Author Contributions

**Conceptualization:** Osamuedeme J. Odiase, Akua O. Gyamerah, Jerry John Nutor, John Koku Awoonor-Williams, Patience A. Afulani.

**Data curation:** Fabian Achana, Monica Getahun, Raymond Aborigo, Jerry John Nutor, Hawa Malechi, Benedicta Arhinful, John Koku Awoonor-Williams, Patience A. Afulani.

**Formal analysis:** Osamuedeme J. Odiase, Akua O. Gyamerah, Fabian Achana, Clara Yang, Sunita Bohara, Patience A. Afulani.

**Funding acquisition:** Jerry John Nutor, Patience A. Afulani.

**Investigation:** Akua O. Gyamerah, Raymond Aborigo, Jerry John Nutor, Hawa Malechi, Benedicta Arhinful, John Koku Awoonor-Williams, Patience A. Afulani.

**Methodology:** Osamuedeme J. Odiase, Akua O. Gyamerah, Raymond Aborigo, Jerry John Nutor, John Koku Awoonor-Williams, Patience A. Afulani.

**Project administration:** Monica Getahun, Raymond Aborigo, Patience A. Afulani.

**Resources:** Jerry John Nutor, Patience A. Afulani.

**Supervision:** Patience A. Afulani.

**Validation:** Osamuedeme J. Odiase, Akua O. Gyamerah, Monica Getahun, Raymond Aborigo, Jerry John Nutor, Hawa Malechi, Benedicta Arhinful, John Koku Awoonor-Williams, Patience A. Afulani.

**Writing – original draft:** Osamuedeme J. Odiase, Akua O. Gyamerah.

**Writing – review & editing:** Osamuedeme J. Odiase, Akua O. Gyamerah, Fabian Achana, Monica Getahun, Clara Yang, Sunita Bohara, Raymond Aborigo, Jerry John Nutor, Hawa Malechi, Benedicta Arhinful, John Koku Awoonor-Williams, Patience A. Afulani.

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
