## [Decision Letter · Decision Letter 0]

7 Dec 2023

PGPH-D-23-02021

Healthcare workers and health system pandemic preparedness: A qualitative study of COVID-19 in Ghana

Dear Dr. Odiase,

Thank you for submitting your manuscript to PLOS Global Public Health. After careful consideration, we feel that it has merit but does not fully meet PLOS Global Public Health’s publication criteria as it currently stands. Therefore, we invite you to submit a revised version of the manuscript that addresses the points raised during the review process.

Please note that we have only been able to secure a single reviewer to assess your manuscript. We are issuing a decision on your manuscript at this point to prevent further delays in the evaluation of your manuscript. Please be aware that the editor who handles your revised manuscript might find it necessary to invite additional reviewers to assess this work once the revised manuscript is submitted. However, we will aim to proceed on the basis of this single review if possible.

The reviewer has commented in particular on the study design, presentation of results, and the discussion. Please ensure you address each of the reviewer's comments when revising your manuscript, some of which can be found in the attached file.

We look forward to receiving your revised manuscript.

Kind regards,

Hugh Cowley

Staff Editor

Journal Requirements:

Additional Editor Comments (if provided):

Reviewers' comments:

Reviewer's Responses to Questions

**Comments to the Author**

1. Does this manuscript meet PLOS Global Public Health’s publication criteria? Is the manuscript technically sound, and do the data support the conclusions? The manuscript must describe methodologically and ethically rigorous research with conclusions that are appropriately drawn based on the data presented.

Reviewer #1: Yes

2. Has the statistical analysis been performed appropriately and rigorously?

Reviewer #1: N/A

3. Have the authors made all data underlying the findings in their manuscript fully available (please refer to the Data Availability Statement at the start of the manuscript PDF file)?

Reviewer #1: Yes

4. Is the manuscript presented in an intelligible fashion and written in standard English?

Reviewer #1: Yes

5. Review Comments to the Author

Reviewer #1: Thank you for your submission of this manuscript. I found that the manuscript will provide important insights of the HCWs in LICs where resource is scarse from non-disaster period. The manuscript will be improved with some additional information and cralifications.

some points for improvements.

1. Information, data clarification and presentations needs to be atteded.

2. data collection methods need to be specified

3. some content clarifications needed some places.

6. PLOS authors have the option to publish the peer review history of their article (what does this mean?). If published, this will include your full peer review and any attached files.

**Do you want your identity to be public for this peer review?** For information about this choice, including consent withdrawal, please see our Privacy Policy.

Reviewer #1: **Yes: **MAYUMI KAKO

---

## [Decision Letter · Decision Letter 1]

11 Mar 2024

PGPH-D-23-02021R1

Healthcare workers and health system pandemic preparedness: A qualitative study of COVID-19 in Ghana

Dear Dr. Odiase,

Thank you for submitting your manuscript to PLOS Global Public Health. After careful consideration, we feel that it has merit but does not fully meet PLOS Global Public Health’s publication criteria as it currently stands. Therefore, we invite you to submit a revised version of the manuscript that addresses the points raised during the review process.

Please pay particular attention to the comments from Reviewers 2 and 3, who both request additional details regarding the sampling methods, rationale, and the suitability of the survey to assess the study aims. Please clearly discuss the study limitations, such as the limited sample size and whether generalization of these findings across the country is valid.

We look forward to receiving your revised manuscript.

Kind regards,

Jennifer Tucker, PhD

Staff Editor

Journal Requirements:

Additional Editor Comments (if provided):

Reviewers' comments:

Reviewer's Responses to Questions

**Comments to the Author**

1. If the authors have adequately addressed your comments raised in a previous round of review and you feel that this manuscript is now acceptable for publication, you may indicate that here to bypass the “Comments to the Author” section, enter your conflict of interest statement in the “Confidential to Editor” section, and submit your "Accept" recommendation.

Reviewer #1: All comments have been addressed

Reviewer #2: (No Response)

Reviewer #3: (No Response)

2. Does this manuscript meet PLOS Global Public Health’s publication criteria? Is the manuscript technically sound, and do the data support the conclusions? The manuscript must describe methodologically and ethically rigorous research with conclusions that are appropriately drawn based on the data presented.

Reviewer #1: Yes

Reviewer #2: Yes

Reviewer #3: No

3. Has the statistical analysis been performed appropriately and rigorously?

Reviewer #1: N/A

Reviewer #2: N/A

Reviewer #3: No

4. Have the authors made all data underlying the findings in their manuscript fully available (please refer to the Data Availability Statement at the start of the manuscript PDF file)?

Reviewer #1: Yes

Reviewer #2: Yes

Reviewer #3: No

5. Is the manuscript presented in an intelligible fashion and written in standard English?

Reviewer #1: Yes

Reviewer #2: Yes

Reviewer #3: Yes

6. Review Comments to the Author

Reviewer #1: Thank you for your work on the comments. the revision of the manuscript addressed the respond of the comments and appropriately responded in the text.

Reviewer #2: Thank you for the opportunity to read this important work. The manuscript was very interesting and presents several key insights from the HCW perspective that will strengthen pandemic preparedness and response. I offer some suggestions the authors may consider:

Major points:

- The methods section would benefit from some brief elaboration on the qualitative approach taken - was this a qualitative descriptive study? Interpretative descriptive? A few sentences on the theoretical underpinnings of the work, or references, would be useful for the reader.

- When presenting the overview of findings, it would be useful for the reader if the authors stated when the level of preparedness survey was conducted (at least the year). This would help the reader contextualize the snapshot that the survey provides within the overall timeline of the COVID-19 pandemic.

- Throughout the text it would be beneficial to put some text that highlights time and when during the pandemic the work was done. For example, it would be useful in the setting to put some time posts in - for example when the first case of COVID-19 was detected in Ghana, when COVID-19 vaccination began in Ghana etc - so that the reader gets a narrative sense of the arc of the pandemic in Ghana from first case to present day and where these interviews fit within that arc. This would also help draw out some key lessons learned from this work for future outbreaks or pandemics.

- I wonder if the authors can do more in the discussion to move beyond the COVID-19 pandemic to make broader points about pandemic preparedness? While the retrospective focus of this work is interesting, I think the work could be strengthened by reflecting on these experiences from the perspective of 2024 and with a consideration of how the work presented here teaches us important lessons to prepare for the next pandemic. This is done a bit in the conclusion but could be done in the discussion as well to better frame the excellent points made in the final paragraph of the study.

Minor points:

- The text switches between ‘health care’ and ‘healthcare.’ I would suggest standardizing one or the other throughout.

Reviewer #3: General comments

The manuscript claims to have assessed healthcare workers and health system panemic preparedness for COVID 19 in Ghana; however, with only 26 respondents interviewed who are responsible for the clinical management of patients and administrators at the health facility level, the study was not designed to provide a valid assessment of the state of Ghana’s health system preparedness. Moreover, further review of the manuscript reveals that it was more about the healthcare workers’ perception of how prepared they were to handle COVID 19 patients at the beginning of the pandemic.

Background

The authors did not provide a good context in previous studies/literature to build the groundwork for how to measure preparedness. Furthermore, the authors did not provide a sufficient background on the Ghanaian health system to allow the reader place the findings in perspective. Authors may wish to access this article by Oppenheim B, Gallivan M, Madhav NK, et al. for ideas on how to assess preparedness

(Oppenheim B, Gallivan M, Madhav NK, et al. Assessing global preparedness for the next pandemic: development and application of an Epidemic Preparedness Index. BMJ Glob Health 2019;4:e001157. doi:10.1136/ bmjgh-2018-001157).

Methods

The details provided in the methodology are not sufficient to address the topic. It is also not clear how issues of bias and confounders were addressed. The sampling methodology employed is not clear. The authors explain that this was part of a survey designed to assess burnout and stress and therefore such a survey may not be designed to assess health systems preparedness.

Results

The methodology and data analysis do not fully support the claims made in the results section. It is not clear the criteria upon whuch the authors arrived at preparedess as presented in Table 1. Also Table 1 has family composition and it is not clear the relationship with preparedness.

7. PLOS authors have the option to publish the peer review history of their article (what does this mean?). If published, this will include your full peer review and any attached files.

**Do you want your identity to be public for this peer review?** For information about this choice, including consent withdrawal, please see our Privacy Policy.

Reviewer #1: **Yes: **MAYUMI KAKO

Reviewer #2: No

Reviewer #3: No

---

## [Editor Report · Decision Letter 2]

24 May 2024

Factors influencing healthcare workers’ and health system preparedness for the COVID-19 pandemic: A qualitative study in Ghana

PGPH-D-23-02021R2

Dear Ms. Odiase,

We are pleased to inform you that your manuscript 'Factors influencing healthcare workers’ and health system preparedness for the COVID-19 pandemic: A qualitative study in Ghana' has been provisionally accepted for publication in PLOS Global Public Health.

Best regards,

Julia Robinson

Executive Editor